# M-polynomial driven machine learning models for predicting physicochemical properties of antibiotics

Xin Li[1]◉, Masoud Ghods◉[2]◉*, Negar Kheirkhahan[2]◉, Jana Shafi[3]

1 Department of Gynecology, Renmin Hospital of Wuhan University, Wuhan, China, 2 Department of Applied Mathematics, Semnan University, Semnan-19111, Iran, 3 Department of Computer Engineering and Information, College of Engineering in Wadi Alddawasir, Prince Sattam Bin Abdulaziz University, Wadi Alddawasir, Saudi Arabia

◉ These authors contributed equally to this work.
* mghods@semnan.ac.ir

## Abstract

Accurate prediction of the physicochemical properties of drug compounds is critical for the development of effective and safe antibiotics. In this study, we employ advanced machine learning techniques to address this challenge, using input data that includes M-Polynomials and various physicochemical descriptors. Three models were implemented: basic Support Vector Regression (SVR-Basic), optimized SVR (SVR-Tuned), and Random Forest (RF), trained on known compounds and tested on previously unseen drug samples to evaluate generalization.

Model performance was comprehensively assessed using $R^2$, MSE, RMSE, and MAE, alongside detailed error and residual analyses to ensure precision and robustness. Furthermore, residual-based metrics such as the Mean Residual (MR), Standard Deviation of Residuals (Std Residual), and Interquartile Range (IQR) of Residuals were employed to provide complementary insights into prediction bias, consistency, and robustness.

By integrating feature importance analysis and ablation studies, the contribution of each molecular descriptor was systematically evaluated, providing deep insights into model stability and the key factors affecting predictive accuracy. Visual comparisons further illustrated the models' behavior on training and test datasets.

The results demonstrate that the proposed approach not only improves predictive accuracy compared to prior studies but also offers a robust and reliable framework for real-world drug development. All models were implemented in Python 3.12.7, highlighting the practical applicability of machine learning in pharmaceutical research.

**Data availability statement:** All relevant data underlying the results presented in this study are publicly available from Figshare at the following DOIs: Test and Training Datasets: Test datasets: https://doi.org/10.6084/m9.figshare.30090193 Training datasets: https://doi.org/10.6084/m9.figshare.30090199 Predictions and Residuals: Predictions MW: https://doi.org/10.6084/m9.figshare.30090160 Predictions PO: https://doi.org/10.6084/m9.figshare.30090169 Predictions COM: https://doi.org/10.6084/m9.figshare.30090226 Residuals summary COM: https://doi.org/10.6084/m9.figshare.30090172, https://doi.org/10.6084/m9.figshare.30090235 Residuals summary MV: https://doi.org/10.6084/m9.figshare.30090175 Residuals summary MW: https://doi.org/10.6084/m9.figshare.30090184 Residuals summary PO: https://doi.org/10.6084/m9.figshare.30090187 Residuals summary (additional): https://doi.org/10.6084/m9.figshare.30090244, https://doi.org/10.6084/m9.figshare.30090271 Ablation Studies and Feature Importance: Ablation study COM: https://doi.org/10.6084/m9.figshare.30090148 Ablation study PO: https://doi.org/10.6084/m9.figshare.30090316 Ablation study MR: https://doi.org/10.6084/m9.figshare.30090334 Feature importance COM: https://doi.org/10.6084/m9.figshare.30090154 Supporting Information Tables: Table S1: https://doi.org/10.6084/m9.figshare.30069574 Table S2: https://doi.org/10.6084/m9.figshare.30069577 Table S3: https://doi.org/10.6084/m9.figshare.30069580 Table S4: https://doi.org/10.6084/m9.figshare.30069583 Table S5: https://doi.org/10.6084/m9.figshare.30069586 Table S6: https://doi.org/10.6084/m9.figshare.30069589 Table S7: https://doi.org/10.6084/m9.figshare.30069598 Table S8: https://doi.org/10.6084/m9.figshare.30069604 Table S9: https://doi.org/10.6084/m9.figshare.30069607 Table S10: https://doi.org/10.6084/m9.figshare.30069610 Code for Predicting Physicochemical Properties: https://doi.org/10.6084/m9.figshare.28790726.

**Funding:** This work was supported by the "Department of Gynaecology, Renmin Hospital of Wuhan University, Wuhan 430060, China" bill to Xin, Li. Xin Li

## Introduction

Bacteria, commonly referred to as microorganisms or microbes, are widely present both within and around the human body. While certain bacterial species are essential for maintaining biological balance, others are responsible for infections such as pharyngitis and urinary tract infections [1]. Antibiotics are critical therapeutic agents used to inhibit or eliminate these pathogenic bacteria, playing a vital role in human medicine, veterinary care, and agriculture [2]. Despite their importance, accurately predicting the physicochemical properties of antibiotics remains a challenge, particularly for new or experimental drug candidates. Addressing this challenge is crucial because precise predictions can accelerate drug development and optimize therapeutic strategies. In recent years, computational approaches such as Quantitative Structure–Property Relationship (QSPR) modeling have emerged as powerful tools in drug discovery and design. QSPR methods correlate molecular structures with physicochemical properties using mathematical and statistical models, enabling the prediction of new compounds' behavior without extensive experimental testing. One fundamental approach for examining the relationship between molecular properties and topological indices is QSPR modeling, which utilizes regression analysis to correlate physicochemical characteristics with topological descriptors. Similarly, QSAR models frequently incorporate these indices to predict biological activity [3,4]. A key concept in this framework is the molecular graph, where atoms are vertices and chemical bonds are edges, analyzed through chemical graph theory. Topological indices (TIs) are widely used descriptors in this context, capturing structural features that relate to molecular properties [5,6]. Several studies have applied TIs in QSPR and QSAR modeling. For instance, S. Kosari analyzed graph structures using the spectral radius and the Zagreb–Estrada index [7], while Kosari et al. proposed bounds for the KG-Sombor index and identified extremal trees achieving those bounds [8]. Beyond these studies, M-polynomials have been introduced as tools to calculate TIs more efficiently and to capture complex molecular features [9,10]. Prior research has shown their application in predicting drug properties for diverse therapeutic areas, including schizophrenia [11], anticancer drugs [12,13], and COVID-19 treatments [14,15]. Machine learning techniques, such as Basic SVR, Tuned SVR, and RF, have further enhanced the predictive power of QSPR models. These methods can capture nonlinear relationships that traditional regression models may overlook, improving prediction accuracy for complex datasets [16–18]. Previous works have demonstrated machine learning applications in QSPR modeling for anxiety treatment drugs and anti-tuberculosis medications [19]. In this study, we extend previous approaches by not only employing advanced machine learning models (SVR, Tuned SVR, and RF) but also focusing on experimental data and previously unseen drug samples used in the treatment of bacterial infections. This allows us to rigorously evaluate the models' predictive accuracy and generalization in real-world settings. By combining mathematical modeling with clinical relevance, our study demonstrates that predictions are both theoretically robust and practically valuable, potentially guiding the development of new antibiotics and optimizing existing therapies. Furthermore, we examine the influence of temperature-related features and nonlinear

was responsible for funding acquisition, conceptualization, investigation, original draft preparation, and review and editing of the manuscript.

**Competing Interests**: The authors have declared that no competing interests exist.

topological indices on model performance, providing deeper insights into the factors that drive molecular behavior [20]. The methodological workflow is illustrated in Fig 1.

## Algorithm for predicting physicochemical properties of antibiotics drugs using machine learning models

### Step 1: Data Preparation and Preprocessing

1. Collect data on M-Polynomials and physicochemical properties of drugs.
2. Handle missing values using median imputation.
3. Detect and remove outliers using the interquartile range (IQR) method.
4. Normalize features using Min–Max scaling to bring values into the range [0, 1].
5. Split the dataset into training (80)

### Step 2: Implementing Machine Learning Models

1. Train the SVR-Basic model using the training data.
2. Optimize and train the SVR-Tuned model.
3. Train the RF model.

### Step 3: Evaluating Model Performance

1. Calculate evaluation metrics including R², MSE, RMSE, MAE, MR, Std, and IQR for each model.
2. Compare model predictions with actual values on the test data.
3. Assess the generalization ability of the models using new drug samples.

### Step 4: Preventing Overfitting

1. Evaluate model performance on both training and test datasets.
2. Generate comparative plots to visualize model performance on both datasets.

### Step 5: Error and Residual Analysis

1. Conduct error distribution analysis to evaluate prediction accuracy.
2. Generate residual plots to assess model consistency and identify patterns or biases.

### Step 6: Analyzing Results and Application in Drug Discovery

1. Assess model accuracy and reliability. model.
2. Analyze the role of machine learning in drug development.
3. Emphasize the importance of using generalizable models for unseen data in real-world applications.

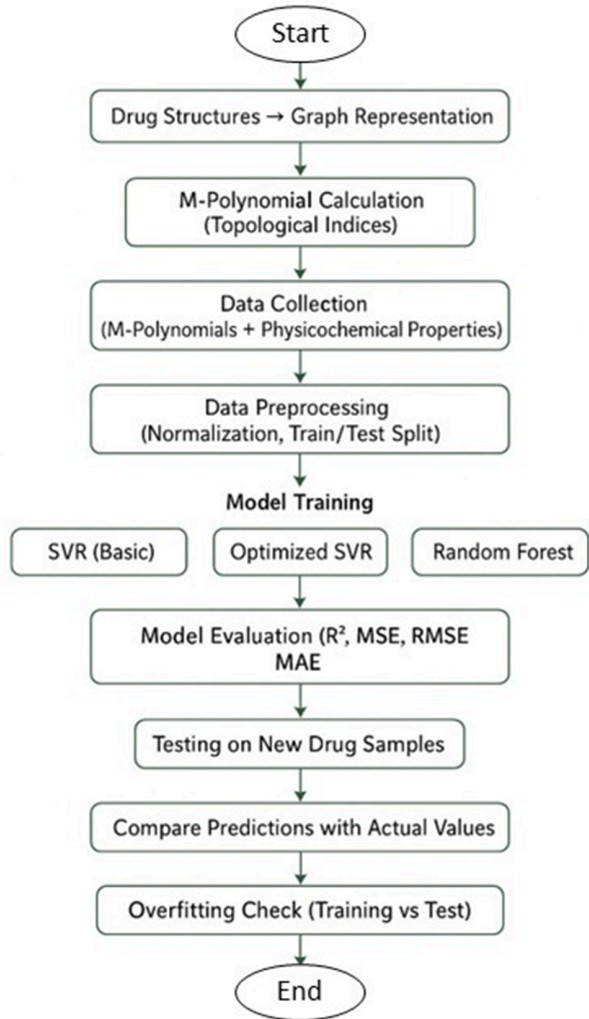

**Fig 1.** **Diagram illustrating the workflow of the employed methodology.** https://doi.org/10.6084/m9.figshare.30443927

## Materials and methods

In this research, antibiotic drugs are represented as basic graph structures. To compute the topological indices of these drug molecules, methods like vertex partitioning, edge partitioning, and several computational techniques have been applied. Our analysis is confined to finite, simple, and connected graphs.

Let $G$ denote a graph with a vertex set $V$ and an edge set $E$. The degree of a vertex $u$, denoted as $d_u$, is defined as the number of vertices adjacent to $u$.

Topological indices are important tools for analyzing molecular and graph structures, and the $M$-polynomial, introduced by Klavžar and Deutsch (2025), enables the calculation of degree-based indices [21]. The topological descriptors related to vertex degree that have been utilized are listed in Table 1.

**Table 1**. Characterization of topological descriptors.

| Topological Descriptors | |
|---|---|
| M-Polynomial | $M(G; x, y) = \sum\limits_{i \leq j} m_{ij}(G) x^i y^j$ |
| First and Second Zagreb Indices | $M_1(G) = \sum\limits_{uv \in E(G)} (d_u + d_v), \quad M_2(G) = \sum\limits_{uv \in E(G)} (d_u \cdot d_v)$ |
| Hyper-Zagreb Index | $HM(G) = \sum\limits_{uv \in E(G)} (d_u + d_v)^2$ |
| Randic Index | $R(G) = \sum\limits_{uv \in E(G)} \sqrt{\frac{1}{d_u \cdot d_v}}$ |
| Harmonic Index | $H(G) = \sum\limits_{uv \in E(G)} \frac{2}{d_u + d_v}$ |
| Sum-Connectivity Index | $S(G) = \sum\limits_{uv \in E(G)} \sqrt{\frac{1}{d_u + d_v}}$ |
| Forgotten Index | $F(G) = \sum\limits_{uv \in E(G)} \left[ (d_u)^2 + (d_v)^2 \right]$ |
| Geometric-Arithmetic Index | $GA(G) = \sum\limits_{uv \in E(G)} \frac{2\sqrt{d_u \cdot d_v}}{d_u + d_v}$ |
| Atomic Bond Connectivity | $ABC(G) = \sum\limits_{uv \in E(G)} \sqrt{\frac{d_u + d_v - 2}{d_u \cdot d_v}}$ |

https://doi.org/10.6084/m9.figshare.29144426

## Methodology and analysis

**SVR-Basic**: Support Vector Regression (SVR) is a robust method for modeling nonlinear relationships by mapping input data into a higher-dimensional feature space using kernel functions such as linear, Gaussian (RBF), and polynomial. SVR minimizes prediction error within a specified tolerance ($\varepsilon$-tube), balancing model complexity through hyperparameters like $C$ (penalty for errors) and $\varepsilon$ (error tolerance).

**SVR-Tuned**: This variant enhances SVR performance by optimizing hyperparameters, including $C$, $\gamma$ (gamma), and $\varepsilon$, using methods such as grid search and random sampling. Hyperparameter tuning allows the model to better capture the underlying data patterns and improve predictive accuracy.

**RF**: RF is an ensemble learning technique that constructs multiple decision trees using random subsets of both data points and features, then aggregates their predictions to enhance accuracy and reduce overfitting. Key hyperparameters include the number of estimators, maximum tree depth, and minimum samples per split. In addition, a brief feature importance analysis is performed for RF, which identifies the features that contribute most to the predictions. This analysis not only aids in interpreting the model's behavior but also guides feature selection for future studies. The performance of all models was evaluated using Mean Squared Error (MSE), Root Mean Squared Error (RMSE), Mean Absolute Error (MAE), $R^2$ score, Mean Residual (MR), Standard Deviation of Residuals (Std), and Interquartile Range (IQR) of Residuals to provide a comprehensive assessment of predictive accuracy.

## Assessment metrics

To evaluate prediction accuracy, four key metrics (models 1 to 7 ) are employed. These metrics measure the precision and efficiency of each model.

$$\text{MSE} = \underbrace{\frac{1}{n} \sum_{i=1}^{n} (y_i - \hat{y}_i)^2}_{\text{Mean Squared Error}} \tag{1}$$

$$RMSE = \underbrace{\sqrt{\frac{1}{n}\sum_{i=1}^{n}(y_i - \hat{y}_i)^2}}_{\text{Root Mean Squared Error}} \tag{2}$$

$$MAE = \underbrace{\frac{1}{n}\sum_{i=1}^{n}|y_i - \hat{y}_i|}_{\text{Mean Absolute Error}} \tag{3}$$

Where $y_i$ is the actual value, $\hat{y}_i$ is the predicted value, and $n$ is the number of samples.

$$R^2 = \underbrace{1 - \frac{\sum_{i=1}^{n}(y_i - \hat{y}_i)^2}{\sum_{i=1}^{n}(y_i - \bar{y}_i)^2}}_{\text{Coefficient of Determination}} \tag{4}$$

Where $\bar{y}$ is the mean of actual values.

$$MR = \underbrace{\frac{1}{n}\sum_{i=1}^{n}(y_i - \hat{y}_i)}_{\text{Mean Residual (MR)}} \tag{5}$$

$$Std = \underbrace{\sqrt{\frac{1}{n-1}\sum_{i=1}^{n}[(y_i - \hat{y}_i) - MR]^2}}_{\text{Standard Deviation of Residuals}} \tag{6}$$

$$IQR = \underbrace{Q_3 - Q_1}_{\text{Interquartile Range}} \tag{7}$$

The best model is the one where the coefficient of determination ($R^2$) is close to 1, as this indicates high accuracy in explaining the variance in the data. Additionally, the MSE, RMSE, and MAE should be as close to 0 as possible, indicating lower error and better performance. Furthermore, residual-based metrics such as the Mean Residual (MR), Standard Deviation of Residuals (Std Residual), and Interquartile Range (IQR) of Residuals provide complementary information about prediction bias, consistency, and robustness. Among the three evaluated models SVR-Basic, SVR-Tuned, and RF a model with $R^2$ close to 1, low error metrics, and low residual variability (small Std Residual and IQR) provides the most accurate and reliable predictions.

**Analysis and comparison of the performance of machine learning models in QSPR evaluation**

In this section, the predictive performance of three machine learning models across five physicochemical properties of antibiotic compounds is evaluated. The analysis includes both numerical metrics and visual assessments to provide a comprehensive understanding of model accuracy and reliability. The models evaluated are **Basic Support Vector Regression (SVR-Basic)**, **Tuned Support Vector Regression (SVR-Tuned)**, and **Random Forest (RF)**.

Nineteen drug compounds were analyzed, and the calculated topological indices are provided in Supplementary Table S1 (S1 Table). Experimental values for the physicochemical properties, sourced from [22], along with predicted values generated using Python, are summarized in various tables. In the main article, Table 2 presents the predicted COM and MR properties, while the remaining properties are available through Supplementary Tables S2 and S3 (S2 Table, S3 Table).

These predictions were compared against experimental data to assess model accuracy. Quantitative metrics such as R², MSE, MAE, RMSE, MR, standard deviation (Std), and interquartile range (IQR) are reported in Tables 3, 4, 5, 6, 7, 8, 9. Additionally, a visual comparison of model performance using selected metrics is shown in Fig 2. The machine learning algorithms used for these predictions are described in detail in the following sections.

**Table 2**. **Prediction results of machine learning models.**

| Comparative Analysis of Machine Learning Models | | | | | | | |
|---|---|---|---|---|---|---|---|
| Chemical Formulas | Actual COM | SVR-Basic | SVR-Tuned | Random-Forest | Actual MR | SVR-Basic | SVR-Tuned | Random-Forest |
| $C_{10}H_{11}N_3O_3S$ | 346 | 815.021 | 345.255 | 572.320 | 62.5 | 113.655 | 62.545 | 85.724 |
| $C_{12}H_{17}N_3O_4S$ | 491 | 813.374 | 489.094 | 572.320 | 72.7 | 112.072 | 72.582 | 85.724 |
| $C_{16}H_{19}N_3O_4S$ | 562 | 811.647 | 562.545 | 567.420 | 83.3 | 111.079 | 83.311 | 85.474 |
| $C_{17}H_{18}FN_3O_3$ | 571 | 811.641 | 570.965 | 571.910 | 89.4 | 111.048 | 89.219 | 89.420 |
| $C_{16}H_{19}N_3O_5S$ | 590 | 811.677 | 590.011 | 593.280 | 89.9 | 111.084 | 90.001 | 89.537 |
| $C_{16}H_{17}N_3O_4S$ | 600 | 811.729 | 598.272 | 588.860 | 91.1 | 111.359 | 91.039 | 90.542 |
| $C_{18}H_{37}N_5O_9$ | 609 | 813.666 | 610.635 | 642.380 | 91.5 | 111.089 | 91.497 | 90.667 |
| $C_{18}H_{20}FN_3O_4$ | 634 | 812.021 | 634.255 | 633.740 | 96.8 | 111.524 | 96.841 | 96.822 |
| $C_{21}H_{43}N_5O_7$ | 636 | 814.610 | 638.490 | 682.530 | 101.8 | 113.254 | 102.007 | 109.748 |
| $C_{17}H_{25}N_3O_5S$ | 679 | 812.075 | 677.997 | 657.060 | 109.0 | 115.386 | 108.990 | 111.078 |
| $C_{21}H_{24}FN_3O_4$ | 727 | 814.128 | 727.602 | 661.320 | 111.7 | 113.392 | 111.713 | 112.014 |
| $C_{22}H_{43}N_5O_{13}$ | 819 | 819.000 | 819.009 | 854.180 | 116.0 | 116.000 | 116.010 | 116.267 |
| $C_{21}H_{39}N_7O_{12}$ | 940 | 820.109 | 938.337 | 942.100 | 121.0 | 117.132 | 121.064 | 121.958 |
| $C_{22}H_{24}N_2O_8$ | 956 | 818.397 | 956.010 | 945.370 | 122.6 | 114.793 | 122.590 | 119.979 |
| $C_{23}H_{27}N_3O_7$ | 971 | 818.824 | 970.372 | 951.370 | 134.9 | 117.559 | 134.890 | 131.051 |
| $C_{38}H_{72}N_2O_{12}$ | 1150 | 821.639 | 1150.220 | 1154.400 | 151.3 | 119.114 | 151.242 | 151.101 |
| $C_{37}H_{67}NO_{13}$ | 1180 | 821.590 | 1179.750 | 1170.300 | 189.2 | 119.964 | 189.211 | 188.692 |
| $C_{38}H_{69}NO_{13}$ | 1190 | 821.454 | 1190.011 | 1177.400 | 194.0 | 119.820 | 194.023 | 191.632 |
| $C_{29}H_{39}N_5O_8$ | 1240 | 821.661 | 1236.959 | 1180.890 | 197.6 | 119.836 | 197.456 | 193.108 |

https://doi.org/10.6084/m9.figshare.29144432

**Table 3**. **Analysis of the performance of machine learning models for drug property prediction based on the R² metric.**

| Models | COM | MR | MV | MW | PO |
|---|---|---|---|---|---|
| SVR-Basic | 0.011136238 | 0.138954133 | -0.03031247 | 0.032089992 | 0.326599397 |
| SVR-Tuned | 0.999976452 | 0.999995247 | 0.990169642 | 0.987028108 | 0.999995603 |
| Random-Forest | 0.944373630 | 0.971139058 | 0.971782379 | 0.964930471 | 0.970835086 |

https://doi.org/10.6084/m9.figshare.29145902

**Table 4**. **Analysis of the performance of advanced ML models for drug property prediction based on the MSE metric.**

| Models | COM | MR | MV | MW | PO |
|---|---|---|---|---|---|
| SVR-Basic | 66971.9789 | 1305.151429 | 20299.08454 | 21632.11019 | 160.2979211 |
| SVR-Tuned | 1.594848005 | 0.007204806 | 193.6764467 | 289.9126973 | 0.001046725 |
| Random-Forest | 3767.362342 | 43.74668242 | 555.9399589 | 783.7793973 | 6.942487158 |

https://doi.org/10.6084/m9.figshare.29145905

**Table 5**. Analysis of the performance of advanced ML models for drug property prediction based on the RMSA metric.

| Models | COM | MR | MV | MW | PO |
|---|---|---|---|---|---|
| SVR-Basic | 258.789449 | 36.12687959 | 142.4748558 | 147.0785851 | 12.66088153 |
| SVR-Tuned | 1.262872917 | 0.084881128 | 13.91676854 | 17.02682288 | 0.03235313 |
| Random-Forest | 61.37884279 | 6.614127488 | 23.57837906 | 27.99606039 | 2.634859988 |

https://doi.org/10.6084/m9.figshare.29145908

**Table 6**. Analysis of the performance of advanced ML models for drug property prediction based on the MAE metric.

| Models | COM | MR | MV | MW | PO |
|---|---|---|---|---|---|
| SVR-Basic | 230.5219014 | 27.24332899 | 94.93726578 | 118.7155829 | 8.939168797 |
| SVR-Tuned | 0.883530924 | 0.058963294 | 10.642956 | 12.37942743 | 0.02324393 |
| Random-Forest | 34.18578947 | 3.464210526 | 16.73752632 | 18.12536842 | 1.523789474 |

https://doi.org/10.6084/m9.figshare.29145911

**Table 7**. Analysis of the performance of advanced ML models for drug property prediction based on the mean residual metric.

| Models | COM | MR | MV | MW | PO |
|---|---|---|---|---|---|
| SVR-Basic | −69.6785 | −46.906 | 25.64194 | −8.12174 | 0.081604 |
| SVR-Tuned | 0.982566 | 0.805262 | −2.07434 | 1.792562 | −0.00106 |
| Random-Forest | −17.8576 | −11.171 | −14.1765 | −10.3209 | −1.31538 |

https://doi.org/10.6084/m9.figshare.30011146

**Table 8**. Analysis of the performance of advanced ML models for drug property prediction based on the std residual metric.

| Models | COM | MR | MV | MW | PO |
|---|---|---|---|---|---|
| SVR-Basic | 428.885818 | 377.0055068 | 155.7821455 | 185.2197339 | 18.00600838 |
| SVR-Tuned | 3.159584389 | 3.038742283 | 17.08657027 | 30.14049248 | 0.063035687 |
| Random-Forest | 238.6656373 | 233.7425558 | 58.53799807 | 72.4473699 | 7.159355684 |

https://doi.org/10.6084/m9.figshare.30011353

**Table 9**. Analysis of the performance of advanced ML models for drug property prediction based on the IQR residual metric.

| Models | COM | MR | MV | MW | PO |
|---|---|---|---|---|---|
| SVR-Basic | 450.3544945 | 276.6445317 | 131.3471582 | 236.5366099 | 12.0964274 |
| SVR-Tuned | 1.4527288 | 0.181922794 | 25.66632321 | 35.83517118 | 0.037285217 |
| Random-Forest | 99.66 | 76.038 | 34.95 | 34.7054 | 2.323 |

https://doi.org/10.6084/m9.figshare.30011356

The results indicate that parameter tuning of the SVR model significantly improves its performance across all evaluation metrics, achieving higher $R^2$ values and lower error rates. While the basic SVR shows weak predictive ability and the Random Forest (RF) performs reasonably well, the tuned SVR consistently outperforms the other models in predicting drug properties.

In Supplementary Table S4 (S4 Table), a simple comparison with a baseline model (e.g., Linear Regression) has been added to provide context for model performance. Data and supplementary materials are available at the following sources:

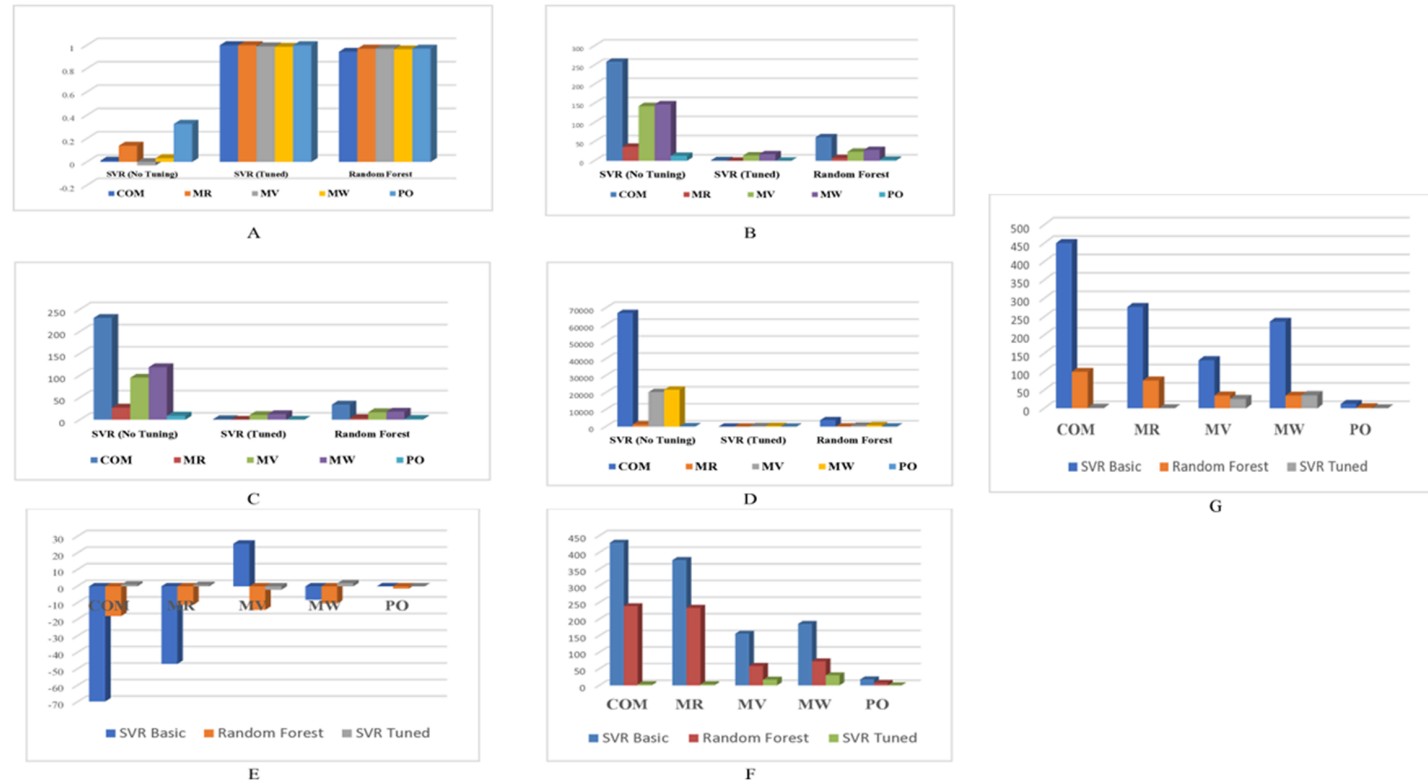

**Fig 2**. **Comparison of machine learning models (basic SVR, tuned SVR, and RF) in predicting physicochemical properties of antibiotic drugs.** $R^2$, MSE, MAE, RMSE, MR, Std, and IQR are shown for training and test datasets to illustrate model accuracy and generalization. **A**: R² comparison of ML models for predicting drug properties. **B**: RMSE comparison of ML models for predicting drugs. **C**: MAE comparison of ML models for predicting drug properties. **D**: MSE comparison of ML models for predicting drug properties. **E**: Mean residual (MR) comparison of ML models for predicting. **F**: Standard deviation (Std) of residuals comparison of ML models for predicting. **G**: Interquartile range (IQR) of residuals comparison of ML models for predicting. Source: DOI:10.6084/m9.figshare.29144435

## Performance analysis of machine learning models using error distributions and residual plots

In this section, we provide a detailed examination of the models' error distributions and residual patterns. These analyses complement the overall performance evaluation presented in the previous section and help to identify the stability and reliability of each model in predicting different physicochemical properties.

To evaluate the performance of different models in predicting physicochemical properties, both error distribution analyses and residual plots were conducted for various features. Fig 3 illustrates the error distribution of different models in predicting the COM property. This figure was generated using histograms combined with Kernel Density Estimation (KDE) curves. The Tuned SVR model shows a narrow and symmetric error distribution centered around zero, indicating high prediction accuracy and low variance. The RF model demonstrates moderate performance with a slightly wider error spread, while the Basic SVR model exhibits the widest error range and the least concentration around zero, reflecting the weakest predictive performance. Fig 4 presents the residual plots of different models for predicting COM. In this plot, residuals are displayed against the predicted values to identify error patterns and potential instabilities. The Tuned SVR model again exhibits a stable and unbiased pattern with residuals evenly dispersed around zero. The RF model follows with moderate stability, while the Basic SVR shows a scattered and less symmetrical distribution of residuals, indicating less reliable predictions. Fig 5 shows the error distribution for the prediction of the MV property. Similar

**Fig 3. Error distributions of the models in predicting COM.** Histograms and KDE plots display the variability and precision of predictions for clear comparison. DOI:10.6084/m9.figshare.29143448

to the observations for COM, the Tuned SVR model achieves superior performance with a sharply peaked distribution near zero. The RF model demonstrates intermediate accuracy, while the Basic SVR again shows broader error dispersion, indicating inferior prediction accuracy. Fig 6 presents the residual plot for MV, which further confirms the trends observed in error distributions. The Tuned SVR maintains a tight and balanced spread of residuals around zero, underscoring its robustness and consistency. The RF model shows slightly greater residual spread but remains reasonably stable. In contrast, the Basic SVR model exhibits high variability and irregular residual patterns, signifying poor stability and less accurate predictions. Overall, these analyses consistently indicate that the Tuned SVR model outperforms the others, providing the most accurate and stable predictions across both COM and MV properties. The RF model ranks second, offering acceptable performance, while the Basic SVR model consistently shows the weakest predictive capacity.

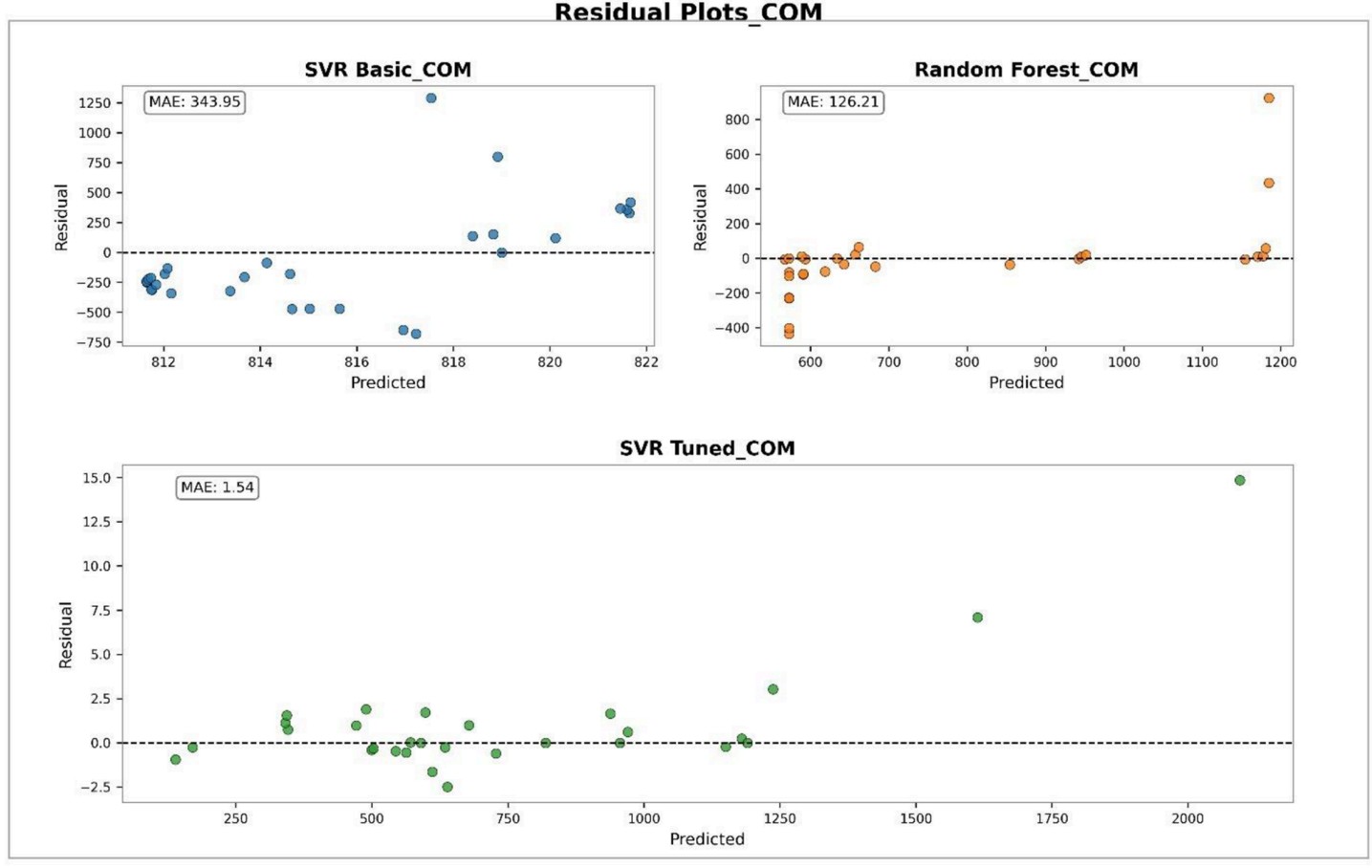

**Fig 4**. **Comparison of residual distributions for different models in predicting COM.** DOI:10.6084/m9.figshare.29143457

## Evaluation of algorithms on test data

In this section, we examine the predictive performance of the three machine learning algorithms on previously unseen test data. This analysis complements the training evaluation and helps assess the models' generalization capability and reliability when applied to new drug compounds.

As mentioned in the previous section, data from nineteen different types of drugs were initially used to train the three machine learning algorithms in Python, aiming to predict their physicochemical properties. Subsequently, ten drug samples were introduced as test data to evaluate model performance.

The comparative results of these predictions for the COM and MR properties are presented as examples in Table 10, while the remaining properties are provided in Supplementary Tables S5 and S6 (S5 Table, S6 Table).

To evaluate the predictive performance of the proposed machine learning models on the test data, several statistical metrics, including R², MSE, RMSE, and MAE, were employed. These metrics provide a comprehensive assessment of both the accuracy and robustness of the models. The comparative results for these metrics are presented individually in Supplementary Tables S7, S8, S9, and S10 (S7 Table, S8 Table, S9 Table, S10 Table), enabling a direct and detailed comparison of the models' predictive capabilities.

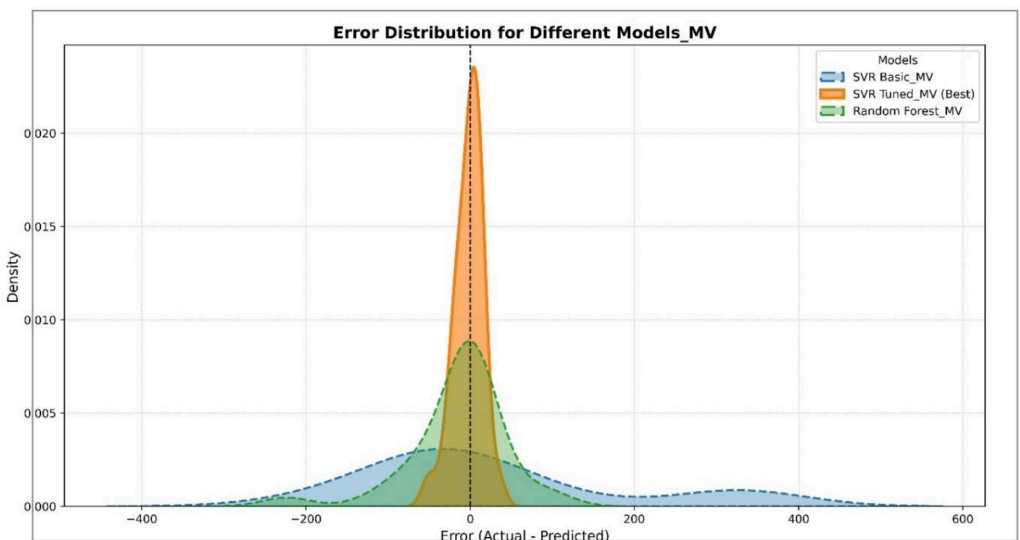

**Fig 5. Error distribution of different models in predicting MV.** DOI:10.6084/m9.figshare.29143463

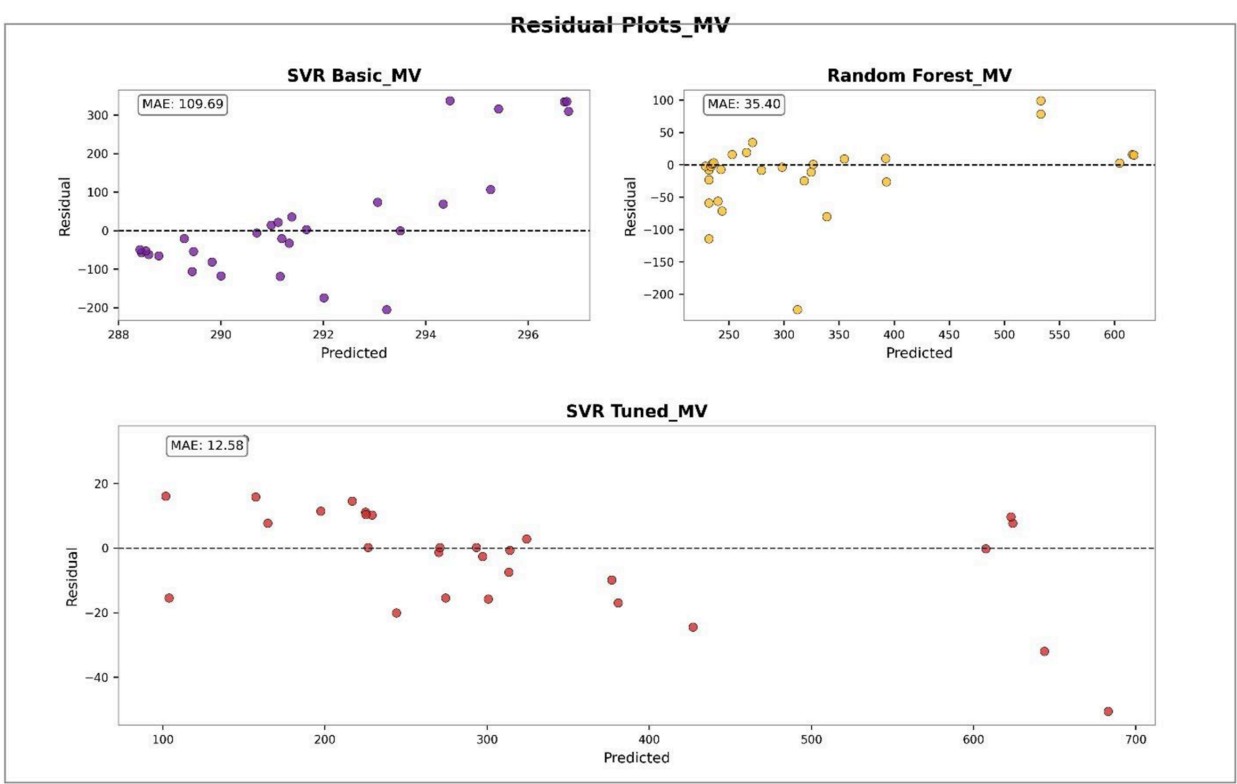

**Fig 6. Residual plot of different models in predicting MV.** DOI:10.6084/m9.figshare.29143472

**Table 10**. Comparison of Actual and Predicted COM and MR Values for Test Drug Samples.

| Chemical Formula | Actual COM | SVR-Basic | SVR-Tuned | RF | Actual MR | SVR-Basic | SVR-Tuned | RF |
|---|---|---|---|---|---|---|---|---|
| $C_{18}H_{33}ClN_2O_5S$ | 138 | 817.2242816 | 138.9393992 | 572.32 | 25.4 | 116.0175305 | 25.62722195 | 85.724 |
| $C_{18}H_{34}N_2O_6S$ | 170 | 816.9597658 | 170.2459142 | 572.32 | 41 | 115.5912842 | 40.91436594 | 85.724 |
| $C_{16}H_{20}FN_3O_4$ | 342 | 814.6535412 | 340.8702735 | 572.32 | 58.9 | 114.1633545 | 59.0159242 | 85.724 |
| $C_{48}H_{62}N_4O_{12}$ | 345 | 815.6395338 | 343.4338781 | 572.32 | 72.6 | 113.1276373 | 72.59749192 | 85.724 |
| $C_{46}H_{62}N_4O_{11}$ | 472 | 812.1512102 | 471.0098558 | 572.32 | 83 | 111.4481342 | 83.09562374 | 88.26 |
| $C_8H_{13}N_3O_4S$ | 499 | 811.7516298 | 499.3900652 | 590.52 | 93.6 | 111.4970079 | 93.54888331 | 90.742 |
| $C_6H_9N_3O_3$ | 502 | 811.7433596 | 502.3230023 | 590.52 | 104.7 | 111.960682 | 104.6144393 | 101.881 |
| $C_{11}H_{12}Cl_2N_2O_5$ | 543 | 811.8377586 | 543.462634 | 618.48 | 107.9 | 112.1718367 | 107.875109 | 103.963 |
| $C_{17}H_{15}FN_6O_3$ | 1620 | 818.9165589 | 1612.898341 | 1184.8 | 213.1 | 117.5990481 | 212.9531734 | 188.686 |
| $C_3H_7O_4P$ | 2110 | 817.5356784 | 2095.145291 | 1184.8 | 222.9 | 116.3854701 | 222.8424053 | 189.67 |

DOI: 10.6084/m9.figshare.29145917

Another notable strength of the proposed approach is the model's stability across the entire range of investigated properties. The performance of the ten test samples is illustrated in Figs 7, 8, 9, and 10, providing a visual analysis of each algorithm's accuracy and reliability in predicting drug properties. The results indicate that the models demonstrated consistent performance on both training and test datasets, implying that they effectively generalized to new data while maintaining high accuracy and robustness. To provide a more concrete representation of the model's performance, three different algorithms were evaluated across four physicochemical properties: MV, PO, COM, and MR. The close alignment

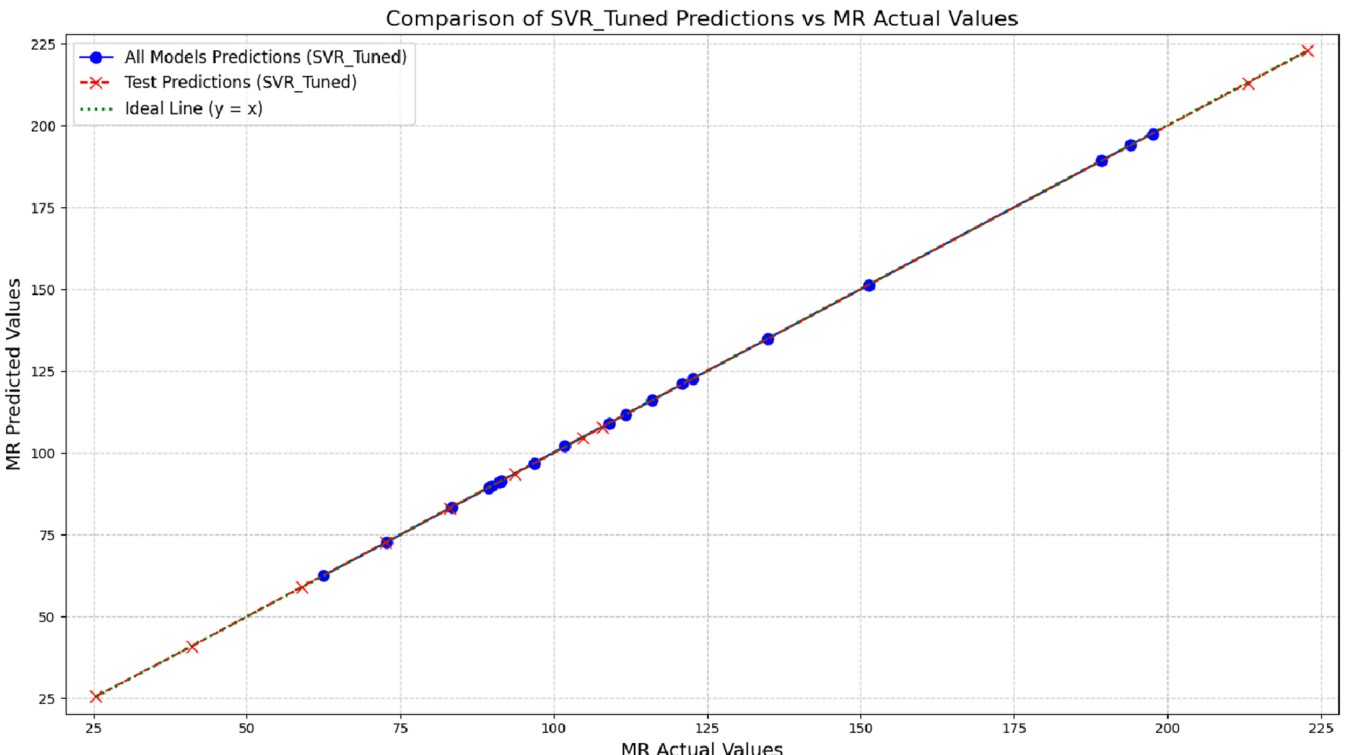

**Fig 7**. Visual comparison of predicted and actual MR values by SVR-Tuned across training and test sets. DOI:10.6084/m9.figshare.29143475

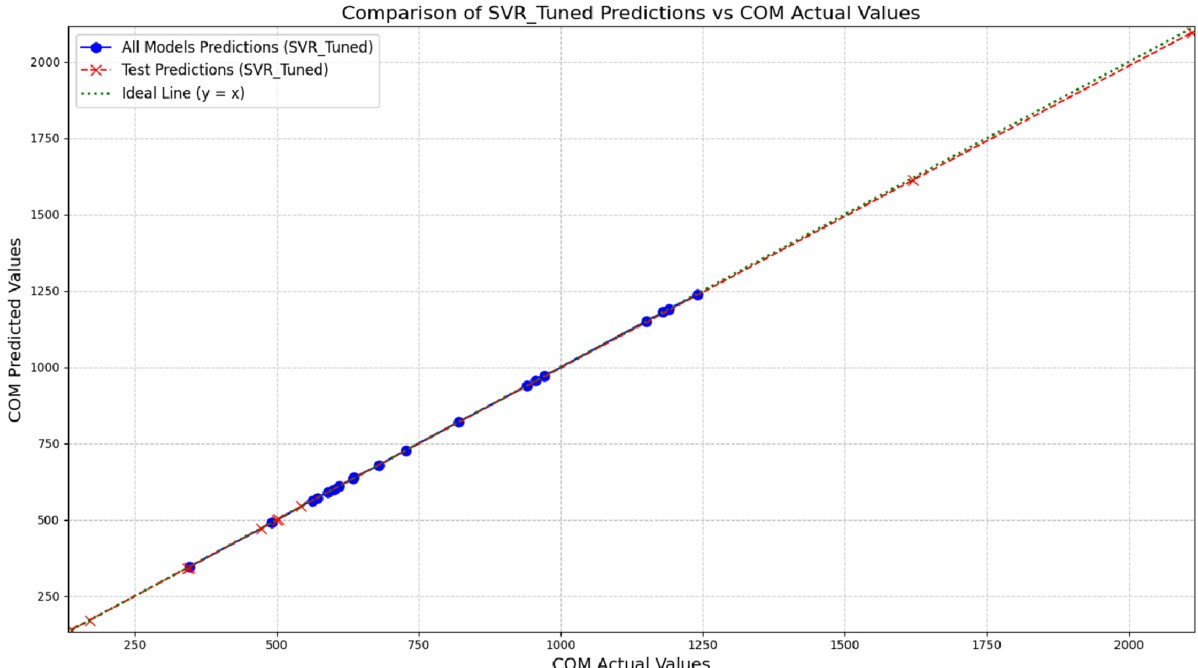

**Fig 8**. **Comparison of predicted and actual COM values using SVR-tuned model on training and test data.** DOI:10.6084/m9.figshare.29143523

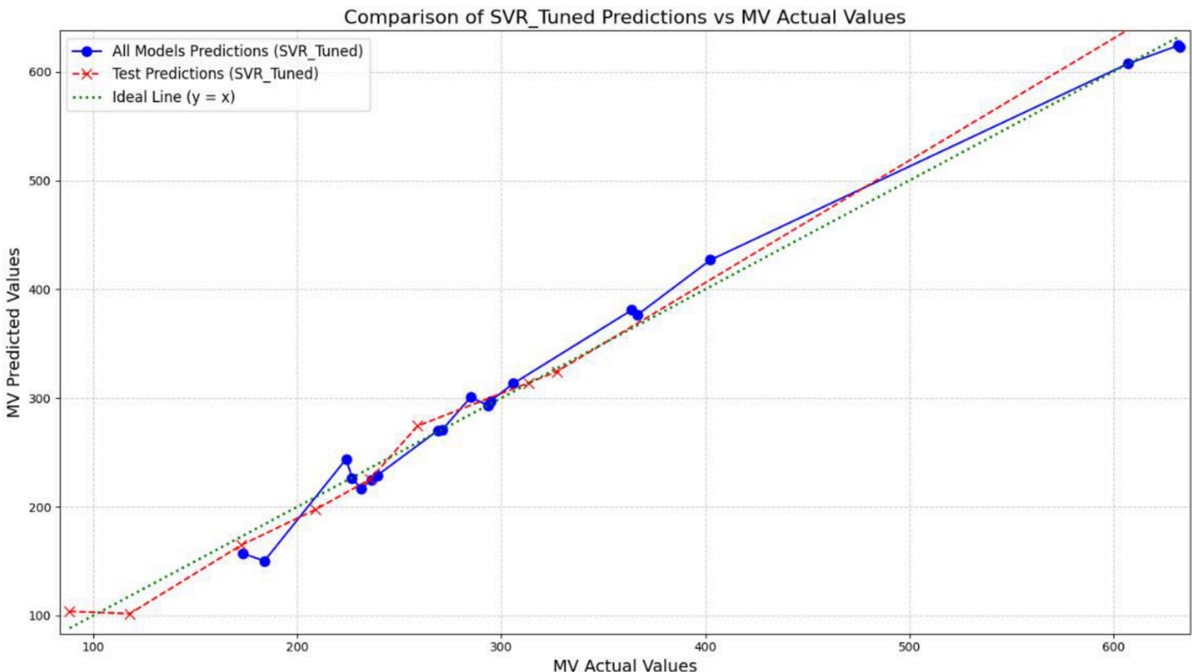

**Fig 9**. **Comparison of SVR-tuned predictions and actual MV values for training and test sets.** DOI:10.6084/m9.figshare.29143526

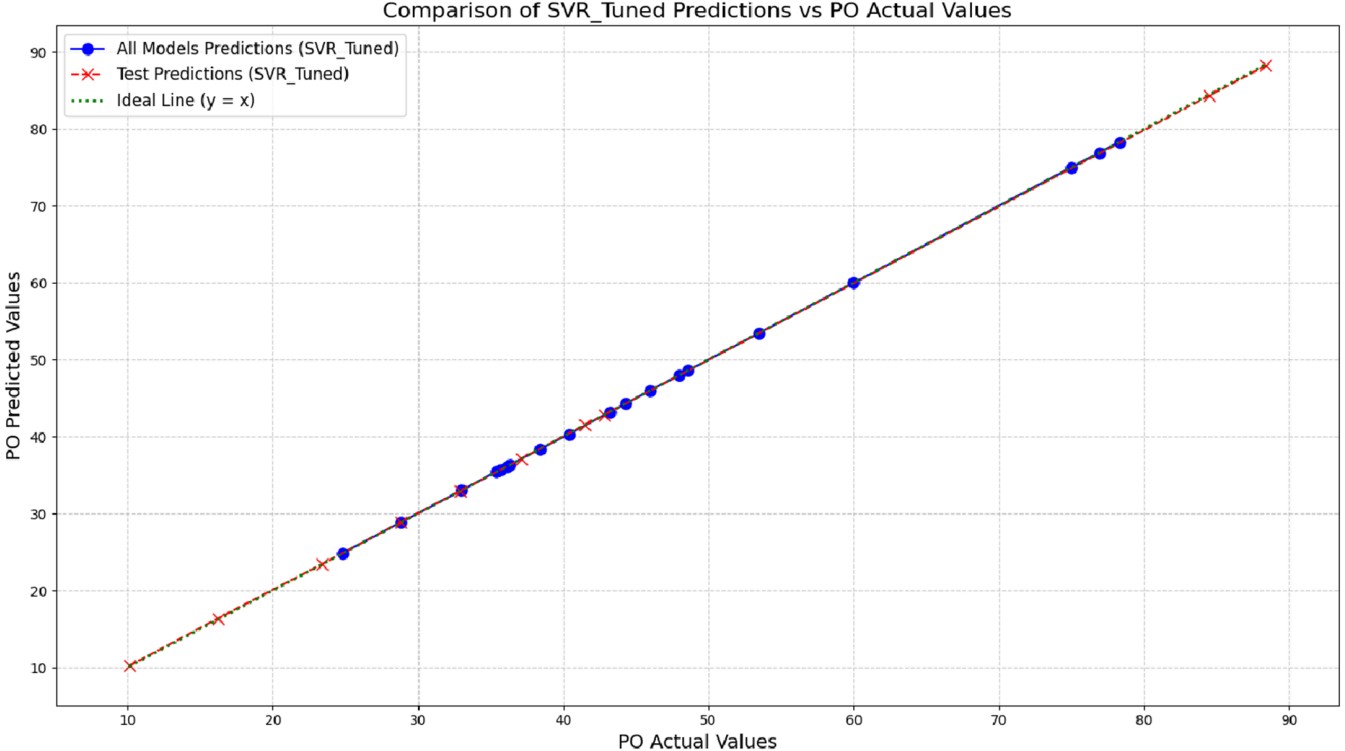

**Fig 10**. **Comparison of SVR-tuned predictions and actual PO values for training and test sets.** DOI:10.6084/m9.figshare.29143529

of the data points (blue representing all data and red representing test data) with the ideal line ($y = x$) illustrates the strong predictive capability of the models. The substantial overlap between training and test predictions indicates consistent performance on unseen data, underscoring strong generalization ability. Moreover, the absence of significant deviations from the ideal line suggests low prediction error and effective learning of the underlying physicochemical patterns. Another notable strength of the proposed approach is the model's stability across the entire range of investigated properties.

## Feature importance

In this section, we present a systematic analysis of feature importance to identify the key molecular descriptors influencing the prediction of five chemical properties. This analysis helps to understand which features contribute most to model accuracy and provides insight into the relative impact of each descriptor on predictive performance. To predict five chemical properties, including COM, MR, PO, MW, and MV, a systematic workflow was followed. First, data cleaning and preparation were conducted: data were imported from an Excel file, duplicate or unnecessary columns were removed, and textual values, such as numbers containing commas, were converted to numeric types. Subsequently, in the feature selection and target variable stage, the values of each property were separated as the target variable, while the remaining descriptors were chosen as input features. The dataset was then split into training (80%) and testing (20%) sets (Train-Test Split). To improve model convergence and standardize the scale of the variables, feature standardization was applied. Following this, Recursive Feature Elimination (RFE) in combination with the RF algorithm was used to select the five most important features for each target property. In the model training phase, three models were employed: Support Vector Regression without hyperparameter tuning (SVR Basic), Support Vector Regression with hyperparameter tuning (SVR Tuned), and RF. Model evaluation was performed for both training and testing sets using four metrics: MAE, RMSE, MSE, and $R^2$.

To further analyze the results, feature importance tables and plots were generated. Specifically, Table 11 presents the comparison of feature importance across different predictive models for COM, MR, MV, MW, and PO, while Fig 11 illustrates the relative importance of features in predicting these chemical indices.

## Ablation study

In this section, we investigate the contribution of individual features to model performance through a systematic ablation study. This analysis quantifies the impact of each molecular descriptor and evaluates model robustness when specific features are excluded.

To examine the contribution of individual features in more detail, an ablation study was conducted. In this study, models were systematically trained and evaluated after removing specific features or groups of features. The results reveal the relative importance of each feature and show how model accuracy is affected when certain descriptors are excluded.

This analysis provides deeper insights into the robustness of the predictive models and highlights the critical role of selected features in forecasting chemical properties. The outcomes are presented in Tables 12–13 and in Figs 12, 13, 14, 15, and 16, which clearly illustrate how the exclusion of each feature influences model performance and identify the most important features contributing to prediction accuracy.

## Conclusions

This study demonstrates the key role of advanced machine learning in accurately predicting the physicochemical properties of drug compounds, which is an important step toward accelerating antibiotic development. Among the evaluated models, SVR-Tuned consistently demonstrated superior performance, achieving substantially higher predictive accuracy and robustness compared to SVR-Basic and RF. Error and residual analyses confirmed the stability of the proposed framework, and evaluations were conducted on both training and unseen test data, clearly demonstrating the models' generalization capability. In addition, feature importance analysis and an ablation study were performed to investigate the contribution of individual molecular descriptors to prediction accuracy. In the feature importance analysis, after data cleaning, preprocessing, and feature scaling, Recursive Feature Elimination (RFE) combined with the RF algorithm was applied to identify the most important descriptors for each target property (COM, MR, MV, MW, and PO). The results indicated that descriptors such as M1(G), PO, GA(G), and COM played a crucial role in predicting various chemical indices, enhancing model interpretability and highlighting the chemical and biological relevance of key features. The ablation study further examined the impact of systematically removing specific features or groups of features on model performance. The exclusion of key descriptors (e.g., M1(G), PO, GA(G), and DE) resulted in a noticeable increase in errors, particularly for SVR-Tuned, which otherwise exhibited the best overall performance. These analyses demonstrated that model accuracy depends not only on algorithmic optimization but also on the careful and meaningful selection of molecular descriptors.

**Table 11. Comparison of feature importance across different prediction models (COM, MR, MV, MW, PO).**

| Fe COM | Im COM | Fe MR | Im MR | Fe MV | Im MV | Fe MW | Im MW | Fe PO | Im PO |
|--------|--------|-------|-------|-------|-------|-------|-------|-------|-------|
| M1(G) | 0.2255 | MV | 0.23464 | PO | 0.25543 | R(G) | 0.22575 | COM | 0.220960 |
| F(G) | 0.2234 | PO | 0.23304 | GA(G) | 0.19725 | GA(G) | 0.21759 | MR | 0.21368 |
| ABC(G) | 0.2179 | M2(G) | 0.19809 | DE | 0.19458 | BP | 0.19732 | SCI(G) | 0.210926 |
| M2(G) | 0.1802 | GA(G) | 0.18747 | IR | 0.17795 | M1(G) | 0.19601 | GA(G) | 0.20473 |
| HM(G) | 0.1527 | COM.1 | 0.14675 | ABC(G) | 0.174779 | H(G) | 0.16331 | M1(G) | 0.149698 |

DOI: 10.6084/m9.figshare.30016408

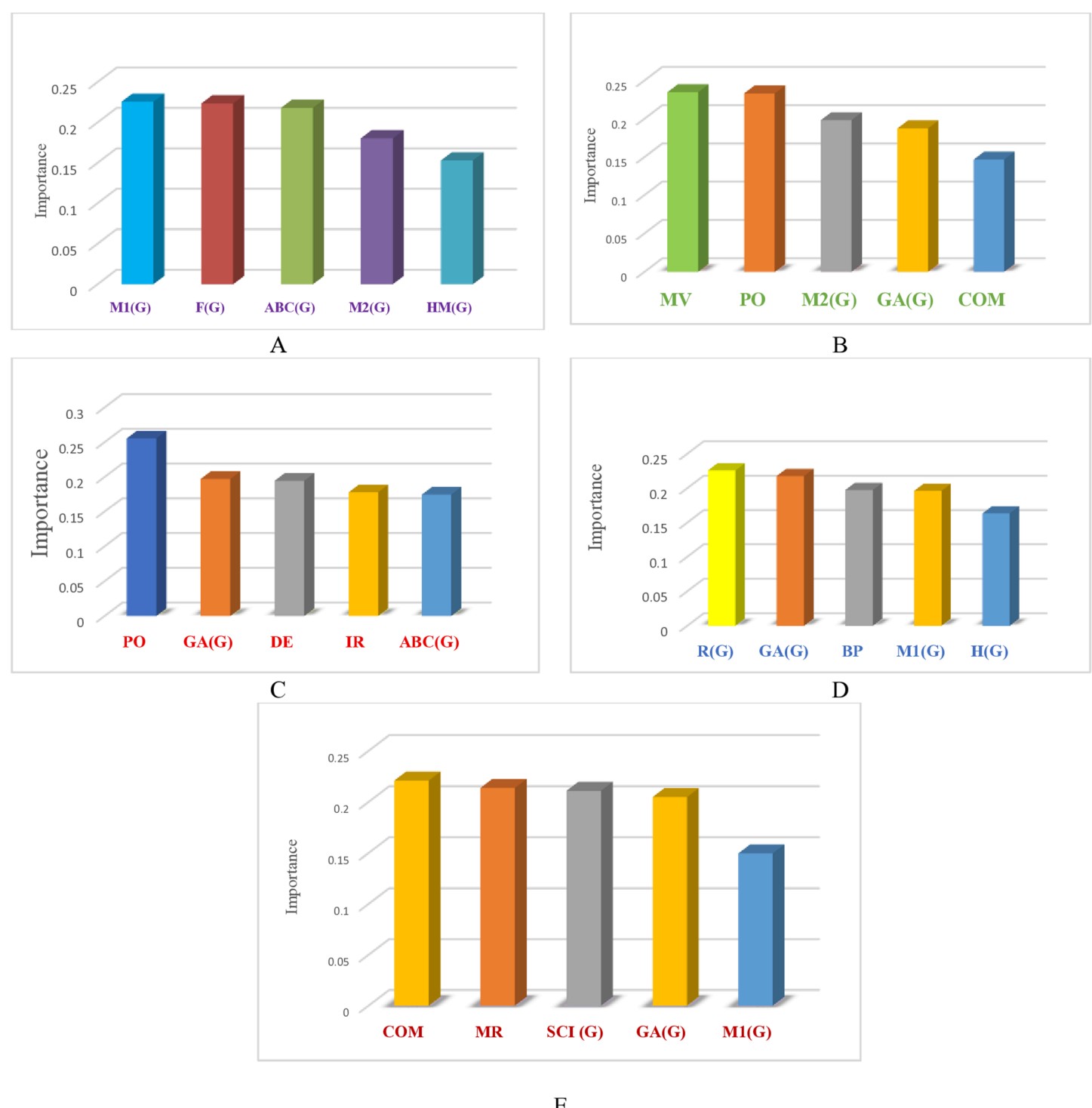

**Fig 11**. **Feature importance in predicting different indicators (COM, MR, MV, MW, PO). A**: Feature importance for predicting COM. **B**: Feature importance for predicting MR. **C**: Feature importance for predicting MV. **D**: Feature importance for predicting MW. **E**: Feature importance for predicting PO. DOI:10.6084/m9.figshare.30068917

**Table 12. Ablation study results: Impact of feature removal on RMSE for COM and MR targets using RF and SVR models.**

| Feature Removed COM | RMSE-RF COM | RMSE-SVR COM | Feature Removed MR | RMSE-RF MR | RMSE-SVR MR |
|---|---|---|---|---|---|
| M1(G) | 136.0237 | 61.3839 | PO | 16.2335 | 3.9021 |
| M2(G) | 137.4284 | 64.5710 | MV | 13.7033 | 0.0955 |
| HM(G) | 137.2593 | 63.1132 | COM | 13.8259 | 0.1215 |
| F(G) | 137.2593 | 63.3622 | M2(G) | 13.8429 | 0.1148 |
| ABC(G) | 136.8545 | 62.3282 | GA(G) | 13.4415 | 0.1169 |

DOI: 10.6084/m9.figshare.30016411

**Table 13. Ablation study results: Impact of feature removal (FR) on RMSE for MV, MW, and PO targets using RF and SVR models.**

| FR MV | RMSE-RF MV | RMSE-SVR MV | FR MW | RMSE-RF MW | RMSE-SVR MW | FR PO | RMSE-RF PO | RMSE-SVR PO |
|---|---|---|---|---|---|---|---|---|
| DE | 43.4112 | 14.8902 | BP | 58.5824 | 23.2947 | COM | 5.3374 | 0.0587 |
| IR | 45.1303 | 25.5183 | M1(G) | 55.4591 | 22.5085 | MR | 6.4896 | 11.9640 |
| PO | 50.2825 | 33.6023 | H(G) | 56.1515 | 24.8892 | M1(G) | 5.3767 | 0.0490 |
| GA(G) | 47.5546 | 18.4053 | R(G) | 55.6691 | 27.9348 | SCI(G) | 5.3747 | 0.0239 |
| ABC(G) | 48.5848 | 19.5190 | GA(G) | 55.3994 | 21.1827 | GA(G) | 5.3747 | 0.0185 |

DOI: 10.6084/m9.figshare.30016417

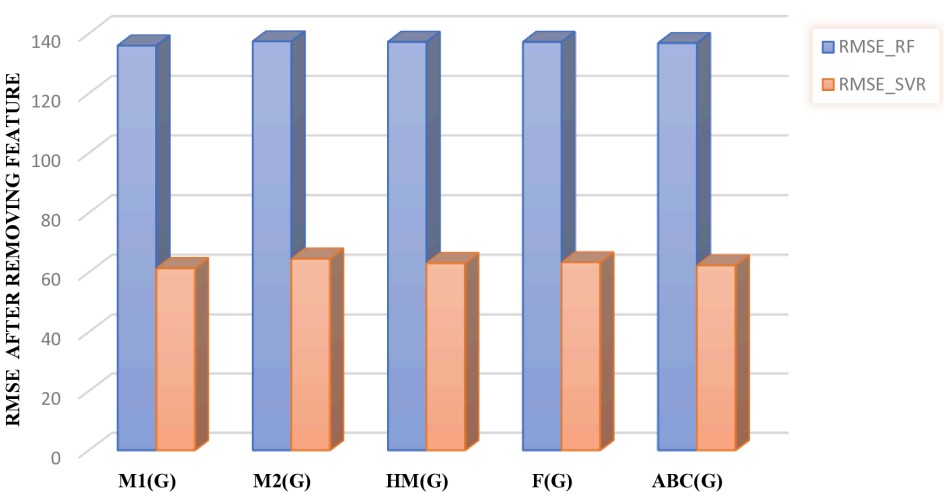

**Fig 12. Ablation analysis of features with respect to RMSE (target: COM).** DOI:10.6084/m9.figshare.30016429

Overall, these findings establish SVR-Tuned as a highly effective, robust, and reliable model for drug property prediction. Feature importance analysis and the ablation study provide deep insights into the contribution of molecular descriptors and the stability of the models, while evaluations on test data confirm strong generalization ability, offering a solid foundation for future applications in computational drug discovery and pharmaceutical research.

The code is available in Supplementary Appendix S1 (S1 Appendix) or via the DOI: Code for predicting physicochemical properties using SVR-Basic, SVR-Tuned, and RF.

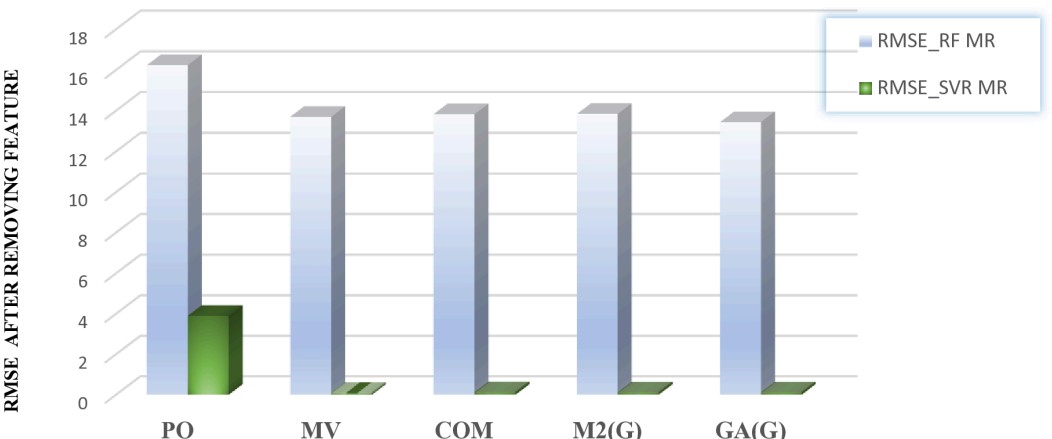

**Fig 13**. **Ablation analysis of features with respect to RMSE (target: MR).** DOI:10.6084/m9.figshare.30016432

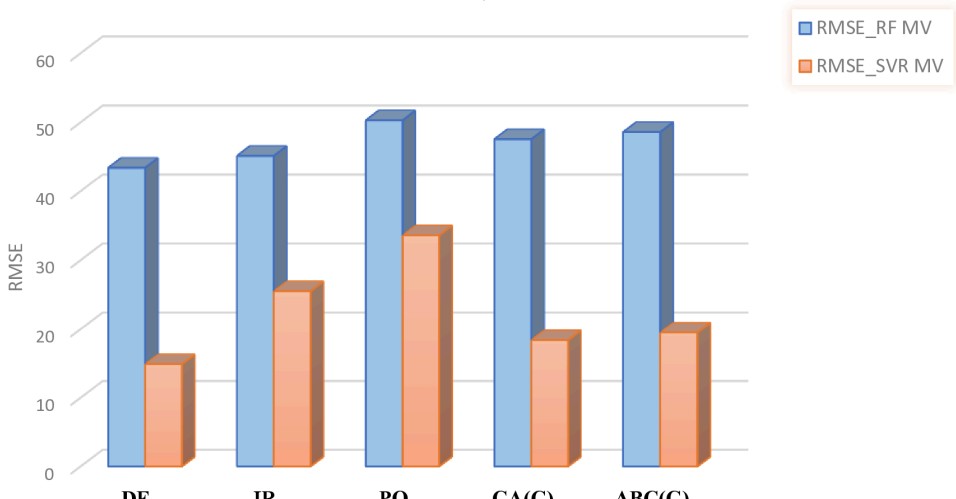

**Fig 14**. **Ablation analysis of features with respect to RMSE (target: MV).** DOI:10.6084/m9.figshare.30016435

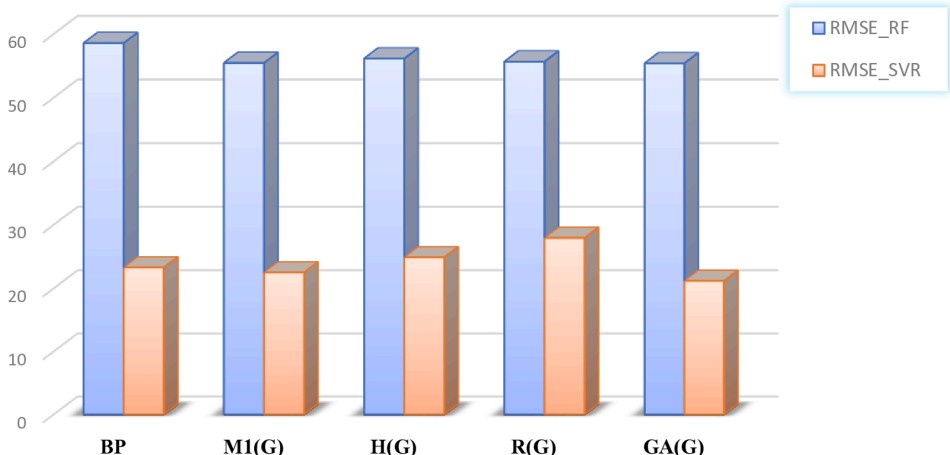

**Fig 15. Ablation analysis of features with respect to RMSE (target: MW).** DOI:10.6084/m9.figshare.30016438

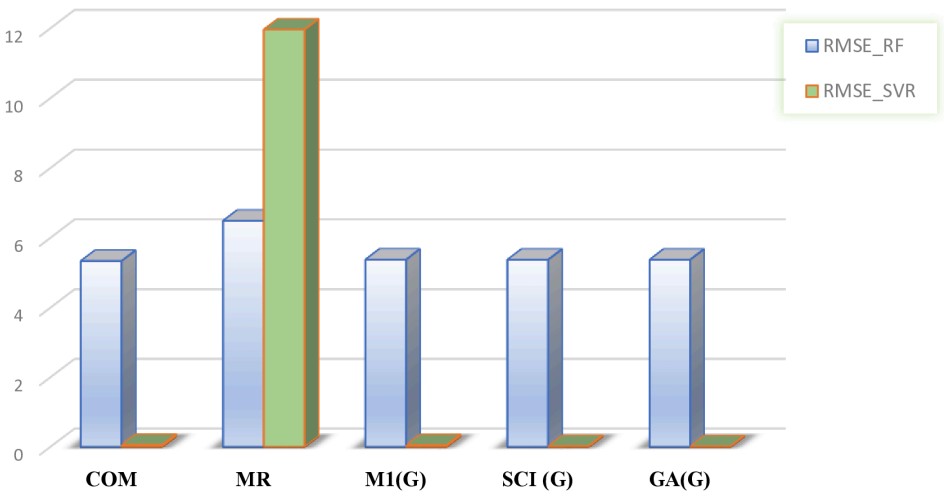

**Fig 16. Ablation analysis of features with respect to RMSE (target: PO).** DOI:10.6084/m9.figshare.30016444

## Supporting information

**S1 Table. Supplementary Table 1.** Available at: https://doi.org/10.6084/m9.figshare.30069574.
(PDF)

**S2 Table. Supplementary Table 2.** Available at: https://doi.org/10.6084/m9.figshare.30069577.
(PDF)

**S3 Table. Supplementary Table 3.** Available at: https://doi.org/10.6084/m9.figshare.30069580.
(PDF)

**S4 Table. Supplementary Table 4.** Available at: https://doi.org/10.6084/m9.figshare.30069583.
(PDF)

**S5 Table. Supplementary Table 5.** Available at: https://doi.org/10.6084/m9.figshare.30069586.
(PDF)

**S6 Table. Supplementary Table 6.** Available at: https://doi.org/10.6084/m9.figshare.30069589.
(PDF)

**S7 Table. Supplementary Table 7.** Available at: https://doi.org/10.6084/m9.figshare.30069598.
(PDF)

**S8 Table. Supplementary Table 8.** Available at: https://doi.org/10.6084/m9.figshare.30069604.
(PDF)

**S9 Table. Supplementary Table 9.** Available at: https://doi.org/10.6084/m9.figshare.30069607.
(PDF)

**S10 Table. Supplementary Table 10.** Available at: https://doi.org/10.6084/m9.figshare.30069610.
(PDF)

**S1 Appendix. Supplementary Appendix 1.** Python code for predicting physicochemical properties using SVR-Basic, SVR-Tuned, and RF. Available at: https://doi.org/10.6084/m9.figshare.28790726.
(PY)

## Acknowledgments

The authors gratefully acknowledge the Deanship of Scientific Research at Prince Sattam bin Abdulaziz University, Al-Kharj, Saudi Arabia, for its support of this research.

## Author contributions

**Conceptualization:** Xin Li, Masoud Ghods, Negar Kheirkhahan, Jana Shafi.

**Data curation:** Xin Li, Masoud Ghods, Negar Kheirkhahan, Jana Shafi.

**Formal analysis:** Xin Li, Masoud Ghods, Negar Kheirkhahan, Jana Shafi.

**Funding acquisition:** Xin Li, Masoud Ghods, Negar Kheirkhahan, Jana Shafi.

**Investigation:** Xin Li, Masoud Ghods, Negar Kheirkhahan, Jana Shafi.

**Methodology:** Xin Li, Masoud Ghods, Negar Kheirkhahan, Jana Shafi.

**Project administration:** Xin Li, Masoud Ghods, Negar Kheirkhahan, Jana Shafi.

**Resources:** Xin Li, Masoud Ghods, Negar Kheirkhahan, Jana Shafi.

**Software:** Xin Li, Masoud Ghods, Negar Kheirkhahan, Jana Shafi.

**Supervision:** Xin Li, Masoud Ghods, Negar Kheirkhahan, Jana Shafi.

**Validation:** Xin Li, Masoud Ghods, Negar Kheirkhahan, Jana Shafi.

**Visualization:** Xin Li, Masoud Ghods, Negar Kheirkhahan, Jana Shafi.

**Writing – original draft:** Xin Li, Masoud Ghods, Negar Kheirkhahan, Jana Shafi.

**Writing – review & editing:** Xin Li, Masoud Ghods, Negar Kheirkhahan, Jana Shafi.

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
