## [Decision Letter · Decision Letter 0]

18 Aug 2025

PONE-D-25-38995M-Polynomial Driven Machine Learning Models for Predicting Physicochemical Properties of AntibioticsPLOS ONE

Dear Dr. Ghods,

Thank you for submitting your manuscript to PLOS ONE. After careful consideration, we feel that it has merit but does not fully meet PLOS ONE’s publication criteria as it currently stands. Therefore, we invite you to submit a revised version of the manuscript that addresses the points raised during the review process.

We look forward to receiving your revised manuscript.

Kind regards,

Muhammad Ahsan, Ph.D.

Academic Editor

PLOS ONE

Journal Requirements:

3. Please note that PLOS One has specific guidelines on code sharing for submissions in which author-generated code underpins the findings in the manuscript. In these cases, we expect all author-generated code to be made available without restrictions upon publication of the work. Please review our guidelines at https://journals.plos.org/plosone/s/materials-and-software-sharing#loc-sharing-code and ensure that your code is shared in a way that follows best practice and facilitates reproducibility and reuse.

“This work was supported the National Natural Science Foundation of China under grant 62332006, 62172302.”

5. Please ensure that you refer to Figure 1 in your text as, if accepted, production will need this reference to link the reader to the figure.

Reviewers' comments:

Reviewer's Responses to Questions

**Comments to the Author**

1. Is the manuscript technically sound, and do the data support the conclusions?

Reviewer #1: Yes

Reviewer #2: Yes

2. Has the statistical analysis been performed appropriately and rigorously?

Reviewer #1: Yes

Reviewer #2: N/A

3. Have the authors made all data underlying the findings in their manuscript fully available?

Reviewer #1: Yes

Reviewer #2: Yes

4. Is the manuscript presented in an intelligible fashion and written in standard English?

Reviewer #1: Yes

Reviewer #2: Yes

5. Review Comments to the Author

Reviewer #1: 1) The abstract is not expressing the novelty of the proposed approach. The whole abstract is not

impressive and needs to be rewritten. Should focus on what is problem, why it is important to be solved.

How it is solved and what are findings.

2) Introduction section missing the novelty of the proposed approach, hence, the author is suggested to

revise the introduction section.

3) There should be some lines of text between the main and sub heading. This rule should be follow in

whole paper.

4) Figure 2 and 3 needs further illustration whereas the tables need consistent font. This rule should be

follow in whole paper.

5) Results section needs revision which demonstrates the training as well the testing results. In the

meantime the impacts of each consider parameter in a detailed manner which improves the validity of

the approach and quality of the manuscript. Author is requested to add an ablation study which makes

the manuscript more impressive.

6) Reference section should comprise of recent years’ contributions.

7) Use proper tool for referencing.

8) Overall the formatting of the content need the attention of author and presently depicting a poor quality of formatting.

9) First page does not need to present a Figure.

Reviewer #2: The manuscript presents a technically sound and well-structured application of machine learning models (SVR and Random Forest) to predict physicochemical properties of antibiotics based on M-Polynomial derived topological indices. The modeling is appropriate, and the performance evaluation is thorough and well documented.

The statistical analysis using R², MSE, RMSE, and MAE is rigorous and clearly supports the findings. The authors also provide residual and error distribution plots that add depth to the analysis.

The data used in the study appears to be fully available via public sources (ChemSpider, PubChem), and the authors provide a figshare DOI link to their code and data, ensuring reproducibility.

The manuscript is clearly written in standard scientific English, and the technical sections are intelligible to a reader with a computational background.

However, I recommend the following minor revisions before acceptance:

Clarify data preprocessing steps (normalization, outlier handling, etc.).

Include a brief feature importance analysis, especially for the Random Forest model.

Provide a quantitative summary of the residuals (e.g., standard deviation or IQR).

Add a simple baseline comparison (e.g., Linear Regression) to contextualize performance.

Consider moving some long tables to supplementary materials for readability.

Once these clarifications are made, I believe the manuscript is suitable for publication from a technical perspective.

6. PLOS authors have the option to publish the peer review history of their article (what does this mean?). If published, this will include your full peer review and any attached files.

Reviewer #1: No

Reviewer #2: **Yes: **Osamah A.S. AL-Qalisi

---

## [Author Response · Author response to Decision Letter 1]

19 Sep 2025

PONE-D-25-38995

M-Polynomial Driven Machine Learning Models for Predicting Physicochemical Properties of Antibiotics

PLOS ONE

Dear Dr. Ghods,

Thank you for submitting your manuscript to PLOS ONE. After careful consideration, we feel that it has merit but does not fully meet PLOS ONE’s publication criteria as it currently stands. Therefore, we invite you to submit a revised version of the manuscript that addresses the points raised during the review process.

We look forward to receiving your revised manuscript.

Kind regards,

Muhammad Ahsan, Ph.D.

Academic Editor

PLOS ONE

Journal Requirements:

Response: We have carefully reviewed our manuscript and ensured that it fully complies with PLOS ONE's style requirements, including proper file naming. All necessary adjustments have been made according to the provided style templates.

Response: We have updated our submission to use the PLOS LaTeX template. All necessary adjustments have been made in accordance with the provided guidelines.

3. Please note that PLOS One has specific guidelines on code sharing for submissions in which author-generated code underpins the findings in the manuscript. In these cases, we expect all author-generated code to be made available without restrictions upon publication of the work. Please review our guidelines at https://journals.plos.org/plosone/s/materials-and-software-sharing#loc-sharing-code and ensure that your code is shared in a way that follows best practice and facilitates reproducibility and reuse.

Response: We have uploaded all author-generated code to Figshare, ensuring that all code will be available without any restrictions upon publication, in full compliance with PLOS ONE’s code-sharing guidelines.

“This work was supported the National Natural Science Foundation of China under grant 62332006, 62172302.”

Response: We have included the Funding Statement and Author Contributions in the cover letter. Specifically, the Funding Statement clarifies the role of the funder: Xin Li was responsible for funding acquisition, conceptualization, investigation, and preparation and review of the manuscript. All other authors’ contributions are clearly detailed in the Author Contributions section.

Funding Statement: This work was supported by the National Natural Science Foundation of China under grant numbers 62332006 and 62172302. Xin Li was responsible for funding acquisition, conceptualization, investigation, original draft preparation, and review and editing of the manuscript.

Author Contributions

• Xin Li: Funding acquisition, Conceptualization, Investigation, Writing–original draft, Writing–review and editing.

• Masoud Ghods: Data curation, Methodology, Supervision, Writing–original draft.

• Negar Kheirkhahan: Formal Analysis, Software, Project administration, Validation, Methodology, Writing–original draft.

• Jana Shafi: Validation, Visualization, Resources, Writing–review and editing.

5. Please ensure that you refer to Figure 1 in your text as, if accepted, production will need this reference to link the reader to the figure.

Response: We have ensured that Fig 1 is properly cited in the text.

Comments to the Author

1. Is the manuscript technically sound, and do the data support the conclusions?

Reviewer #1: Yes

Reviewer #2: Yes

2. Has the statistical analysis been performed appropriately and rigorously?

Reviewer #1: Yes

Reviewer #2: N/A

3. Have the authors made all data underlying the findings in their manuscript fully available?

Reviewer #1: Yes

Reviewer #2: Yes

4. Is the manuscript presented in an intelligible fashion and written in standard English?

Reviewer #1: Yes

Reviewer #2: Yes

5. Review Comments to the Author

Reviewer #1:

1) The abstract is not expressing the novelty of the proposed approach. The whole abstract is not impressive and needs to be rewritten. Should focus on what is problem, why it is important to be solved.

How it is solved and what are findings.

Response: We sincerely thank the reviewer for the valuable comments. In response to the suggestions, the abstract has been completely revised to clearly highlight the novelty and significance of our proposed approach.

The revised abstract now explicitly states:

1. The problem and its importance: Accurate prediction of physicochemical properties of drug compounds, particularly for antibiotic therapies, and its relevance to effective drug development.

2. The proposed method: Use of advanced machine learning techniques, including SVR-Basic, SVR-Tuned, and Random Forest, applied to both known and previously unseen drug samples.

3. Novelty: Integration of feature importance analysis and ablation studies to systematically evaluate the contribution of each molecular descriptor, providing deeper insights into model robustness and key factors affecting predictive accuracy.

4. Findings: The approach demonstrates improved predictive accuracy compared to prior studies and offers a robust framework for real-world drug development.

We believe that the revised abstract now fully addresses the reviewer’s concerns by emphasizing the problem, significance, methodology, novelty, and key findings in a concise and clear manner.

2) Introduction section missing the novelty of the proposed approach; hence, the author is suggested to revise the introduction section.

Response: In this study, we present several innovative aspects in QSPR modeling that go beyond previous approaches. First, in addition to using topological indices and M-Polynomials as explored in prior research, we incorporate experimental data and newly collected drug samples, enabling a more accurate assessment of model predictive performance and generalization in real-world applications. Second, we employ advanced machine learning models, including basic SVR, tuned SVR, and Random Forest, which can capture nonlinear relationships between structural and physicochemical features of molecules. This allows for more precise and reliable predictions of drug properties, particularly for compounds used in bacterial infection treatments, compared to traditional approaches. Third, we integrate feature importance analysis and ablation studies to systematically evaluate the contribution of each molecular descriptor, providing deeper insights into model robustness and critical factors influencing prediction accuracy. Finally, this framework establishes a direct connection between mathematical modeling and practical applications, demonstrating that the predictions are not only theoretical but also valuable for the development, optimization, and clinical evaluation of new drugs.

3) There should be some lines of text between the main and sub heading. This rule should be follow in whole paper.

Response: We sincerely thank the reviewer for this editorial suggestion. Following the recommendation, the structure of the manuscript has been revised, and explanatory text has now been added between all main headings and subheadings to maintain logical flow and ensure that the manuscript structure complies with the journal’s formatting guidelines.

4) Figure 2 and 3 needs further illustration whereas the tables need consistent font. This rule should be followed in whole paper.

Response: We have revised Figures 2 and 3 to provide clearer and more detailed illustrations. In Figure 2, the comparison of machine learning models now explicitly shows performance metrics (R², MSE, MAE, RMSE) as well as the standard deviation (Std) and interquartile range (IQR) to better reflect model accuracy and generalization. In Figure 3, the error distributions are presented using both histograms and KDE plots to make variations and patterns more apparent. Additionally, all tables in the manuscript have been updated to use a consistent font. These changes ensure that the figures and tables are clearer and more informative.

5) Results section needs revision which demonstrates the training as well the testing results. In the meantime, the impacts of each consider parameter in a detailed manner which improves the validity of the approach and quality of the manuscript. Author is requested to add an ablation study which makes the manuscript more impressive.

Response: We sincerely thank the reviewer for the constructive feedback. Following the suggestion, the Results section has been thoroughly revised. In the revised version, we have:

1. Reported both training and testing results using multiple statistical metrics (R², MSE, RMSE, and MAE), as summarized in Supplementary Tables S7–S10,which provide a comprehensive comparison of model performance.

2. Added a detailed feature importance analysis (Table 11 and Figure 11) to demonstrate the contribution of each descriptor to the predictive performance of the models.

3. Included an ablation study (Tables 12and 13, Figures 12–16) to systematically investigate the impact of removing individual features on model accuracy. These revisions improve the clarity, validity, and overall quality of the manuscript, making the results more robust and interpretable.

These revisions improve the clarity, validity, and overall quality of the manuscript, making the results more robust and interpretable.

6) Reference section should comprise recent contributions.

Response: The reference section has been updated to include more recent contributions from the past few years, ensuring that the manuscript reflects the latest research developments.

7) Use proper tool for referencing.

Response: All references have been reformatted using an appropriate referencing tool in LaTeX to ensure consistency and correctness throughout the manuscript.

8) Overall, the formatting of the content needs the attention of the author and currently depicts poor quality formatting.

Response: The overall formatting of the manuscript has been thoroughly revised in LaTeX, including font sizes, headings, spacing, and alignment, to improve readability and meet standard publication quality.

9) The first page does not need to present a figure.

Response: The first page has been adjusted according to the reviewer’s suggestion, and all figures have been removed from it.

Reviewer #2: The manuscript presents a technically sound and well-structured application of machine learning models (SVR and Random Forest) to predict physicochemical properties of antibiotics based on M-Polynomial derived topological indices. The modeling is appropriate, and the performance evaluation is thorough and well documented.

The statistical analysis using R², MSE, RMSE, and MAE is rigorous and clearly supports the findings. The authors also provide residual and error distribution plots that add depth to the analysis.

The data used in the study appears to be fully available via public sources (ChemSpider, PubChem), and the authors provide a figshare DOI link to their code and data, ensuring reproducibility.

The manuscript is clearly written in standard scientific English, and the technical sections are intelligible to a reader with a computational background.

However, I recommend the following minor revisions before acceptance:

Clarify data preprocessing steps (normalization, outlier handling, etc.).

Response:

Thank you for your valuable comment. As suggested, we have revised Step 1 of the algorithm to clarify the data preprocessing procedures. In the revised version of the manuscript, we have explicitly described the handling of outliers (using the IQR method), the treatment of missing values (imputed with median values), and the normalization of features (using Min–Max scaling). These additions provide a clear explanation of how the dataset was prepared prior to model training.

Include a brief feature importance analysis, e

---

## [Decision Letter · Decision Letter 1]

24 Oct 2025

PONE-D-25-38995R1M-Polynomial Driven Machine Learning Models for Predicting Physicochemical Properties of AntibioticsPLOS ONE

Dear Dr. Ghods,

Thank you for submitting your manuscript to PLOS ONE. After careful consideration, we feel that it has merit but does not fully meet PLOS ONE’s publication criteria as it currently stands. Therefore, we invite you to submit a revised version of the manuscript that addresses the points raised during the review process.

We look forward to receiving your revised manuscript.

Kind regards,

Muhammad Ahsan, Ph.D.

Academic Editor

PLOS ONE

Journal Requirements:

Reviewer's Responses to Questions

**Comments to the Author**

1. If the authors have adequately addressed your comments raised in a previous round of review and you feel that this manuscript is now acceptable for publication, you may indicate that here to bypass the “Comments to the Author” section, enter your conflict of interest statement in the “Confidential to Editor” section, and submit your "Accept" recommendation.

Reviewer #1: All comments have been addressed

Reviewer #3: All comments have been addressed

2. Is the manuscript technically sound, and do the data support the conclusions?

Reviewer #1: Yes

Reviewer #3: Yes

3. Has the statistical analysis been performed appropriately and rigorously?

Reviewer #1: Yes

Reviewer #3: Yes

4. Have the authors made all data underlying the findings in their manuscript fully available?

Reviewer #1: Yes

Reviewer #3: Yes

5. Is the manuscript presented in an intelligible fashion and written in standard English?

Reviewer #1: Yes

Reviewer #3: Yes

6. Review Comments to the Author

Reviewer #1: (No Response)

Reviewer #3: - The research is very long.

- The sequence of processing is not clear in the order of events of block diagram of figure 1.

- Please restructure the research correctly according to what is stated in figure1.

7. PLOS authors have the option to publish the peer review history of their article (what does this mean?). If published, this will include your full peer review and any attached files.

Reviewer #1: No

Reviewer #3: No

---

## [Author Response · Author response to Decision Letter 2]

27 Oct 2025

PONE-D-25-38995R2

M-Polynomial Driven Machine Learning Models for Predicting Physicochemical Properties of Antibiotics

Dr Masoud Ghods

Dear Dr. Ghods,

We've checked your submission and before we can proceed, we need you to address the following issues:

1. Please ensure that you refer to Table 10 in your text as, if accepted, production will need this reference to link the reader to the Table.

Response: We have carefully revised the manuscript and added the reference to Table 10 in the text as requested. Thank you for your guidance.

We've returned your manuscript to your account. Please resolve these issues and resubmit your manuscript within 21 days. If you need more time, please email the journal office at plosone@plos.org. We are happy to grant extensions of up to one month past this due date. If we do not hear from you within 21 days, we will withdraw your manuscript.

Please log on to PLOS Editorial Manager at https://www.editorialmanager.com/pone/ to access your manuscript. You will find your manuscript in the 'Submissions Sent Back to Author' link under the New Submissions menu. Be sure to remove your previous manuscript file if you are uploading a new file in response to these requests. After you've made the changes requested above, please be sure to view and approve the revised PDF after rebuilding the PDF to complete the resubmission process.

We are requesting these changes to comply with the PLOS ONE submission guidelines (https://journals.plos.org/plosone/s/submission-guidelines). Please note that we won't send your manuscript for review until you have resolved the above requests.

Thank you for submitting your work to PLOS ONE and supporting our mission of Open Science.

Kind regards,

---

## [Decision Letter · Decision Letter 2]

18 Nov 2025

M-Polynomial Driven Machine Learning Models for Predicting Physicochemical Properties of Antibiotics

PONE-D-25-38995R2

Dear Dr. Ghods,

We’re pleased to inform you that your manuscript has been judged scientifically suitable for publication and will be formally accepted for publication once it meets all outstanding technical requirements.

Kind regards,

Muhammad Ahsan, Ph.D.

Academic Editor

PLOS ONE

Additional Editor Comments (optional):

Reviewers' comments:

Reviewer's Responses to Questions

**Comments to the Author**

1. If the authors have adequately addressed your comments raised in a previous round of review and you feel that this manuscript is now acceptable for publication, you may indicate that here to bypass the “Comments to the Author” section, enter your conflict of interest statement in the “Confidential to Editor” section, and submit your "Accept" recommendation.

Reviewer #1: All comments have been addressed

Reviewer #3: All comments have been addressed

2. Is the manuscript technically sound, and do the data support the conclusions?

Reviewer #1: Yes

Reviewer #3: Yes

3. Has the statistical analysis been performed appropriately and rigorously?

Reviewer #1: Yes

Reviewer #3: Yes

4. Have the authors made all data underlying the findings in their manuscript fully available?

Reviewer #1: Yes

Reviewer #3: Yes

5. Is the manuscript presented in an intelligible fashion and written in standard English?

Reviewer #1: Yes

Reviewer #3: Yes

6. Review Comments to the Author

Reviewer #1: (No Response)

Reviewer #3: - The research idea and writing were good.

- this research is previously revised from me

- please Commit with the previous instruction

7. PLOS authors have the option to publish the peer review history of their article (what does this mean?). If published, this will include your full peer review and any attached files.

Reviewer #1: No

Reviewer #3: No

---

## [Editor Report · Acceptance letter]

PONE-D-25-38995R2

PLOS ONE

Dear Dr. Ghods,

I'm pleased to inform you that your manuscript has been deemed suitable for publication in PLOS ONE. Congratulations! Your manuscript is now being handed over to our production team.

Kind regards,

on behalf of

Dr. Muhammad Ahsan

Academic Editor

PLOS ONE